# Virulent Phages Isolated from a Smear-Ripened Cheese Are Also Detected in Reservoirs of the Cheese Factory

**DOI:** 10.3390/v14081620

**Published:** 2022-07-25

**Authors:** Thomas Paillet, Julien Lossouarn, Clarisse Figueroa, Cédric Midoux, Olivier Rué, Marie-Agnès Petit, Eric Dugat-Bony

**Affiliations:** 1Université Paris-Saclay, INRAE, AgroParisTech, UMR SayFood, 91120 Palaiseau, France; thomas.paillet@inrae.fr (T.P.); clarisse.figueroa4@gmail.com (C.F.); 2Université Paris-Saclay, INRAE, AgroParisTech, Micalis Institute, 78352 Jouy-en-Josas, France; julien.lossouarn@inrae.fr (J.L.); marie-agnes.petit@inrae.fr (M.-A.P.); 3Université Paris-Saclay, INRAE, MaIAGE, 78350 Jouy-en-Josas, France; cedric.midoux@inrae.fr (C.M.); olivier.rue@inrae.fr (O.R.); 4Université Paris-Saclay, INRAE, BioinfOmics, MIGALE Bioinformatics Facility, 78350 Jouy-en-Josas, France; 5Université Paris-Saclay, INRAE, PROSE, 92761 Antony, France

**Keywords:** smear-ripened cheese, virulent phages, rind bacteria, phage reservoirs, viral genomics

## Abstract

Smear-ripened cheeses host complex microbial communities that play a crucial role in the ripening process. Although bacteriophages have been frequently isolated from dairy products, their diversity and ecological role in such this type of cheese remain underexplored. In order to fill this gap, the main objective of this study was to isolate and characterize bacteriophages from the rind of a smear-ripened cheese. Thus, viral particles extracted from the cheese rind were tested through a spot assay against a collection of bacteria isolated from the same cheese and identified by sequencing the full-length small subunit ribosomal RNA gene. In total, five virulent bacteriophages infecting *Brevibacterium aurantiacum*, *Glutamicibacter arilaitensis*, *Leuconostoc falkenbergense* and *Psychrobacter aquimaris* species were obtained. All exhibit a narrow host range, being only able to infect a few cheese-rind isolates within the same species. The complete genome of each phage was sequenced using both Nanopore and Illumina technologies, assembled and annotated. A sequence comparison with known phages revealed that four of them may represent at least new genera. The distribution of the five virulent phages into the dairy-plant environment was also investigated by PCR, and three potential reservoirs were identified. This work provides new knowledge on the cheese rind viral community and an overview of the distribution of phages within a cheese factory.

## 1. Introduction

Cheese is a fermented food hosting a complex microbial ecosystem, comprising bacteria (mainly *Firmicutes*, *Actinobacteria* and *Proteobacteria*), yeasts and molds in varying proportions [1]. These microorganisms can naturally originate from raw milk or colonize the facility environment. They are thus commonly referred to as the “house” microbiota, which is specific to each dairy plant [2]. They can also be intentionally added during the cheese production process via the use of commercial starters, ripening cultures or through back slopping procedures [3,4]. The controlled succession of these microorganisms all along the production process is key in obtaining a final product meeting the expectations of the consumer in terms of visual appearance, organoleptic qualities and safety.

In dairy plants, equipment is carefully washed to avoid the formation of biofilms and, moving forward, logic is widely applied to avoid cross-contaminations. At the industrial scale, raw milk is often pasteurized at the beginning of the process [5] to eliminate undesirable microorganisms before starter cultures’ inoculation. Acidifying starter cultures, composed of lactic acid bacteria (LAB), such as *Lactococcus lactis*, *Streptococcus thermophilus* or *Lactobacillus* species, provides a fast acidification of the milk that helps the coagulation process and reduces the growth of acid-sensitive bacteria [6], in particular pathogens and spoilage microorganisms. They also participate in the overall degradation of milk constituents and production of aroma compounds, e.g., through their proteolytic activities and amino acids’ catabolism [7]. The combination of all the technological factors and microbial inhibition activities presented above, known as the hurdle technology [8], ensures achieving a safe product with acceptable shelf life [3]. However, a recurrent agent is still often beyond the control of these measures: the bacteriophages. Bacteriophages, or phages, are viruses infecting bacteria to replicate and represent a key player in the dynamics of many microbial ecosystems [9,10,11]. In fermented foods, and dairy products in particular, phages are frequently isolated after a fermentation failure [12]. As many phages are able to infect acidifying starter cultures (e.g., *Lactococcus lactis*, *Streptococcus thermophilus*, *Lactobacillus delbrueckii*) [13], their detrimental impact on milk acidification is very well-described in the literature [14,15,16]. Phage issues concern every size of plants, every kind of dairy product [17,18,19,20], and may result in important economic losses.

Among dairy products, smear-ripened cheeses are of special interest because of their singular production process, involving several washing steps with a saline solution sometimes mixed with alcoholic beverages (wine, beer or liquors). Smear-ripened cheeses possess a typical viscous, red-orange smear on their surface which is mainly composed of bacteria and yeasts [21,22]. The bacteria observed in the rind of such a type of cheese are diverse and comprise coryneform bacteria, staphylococci and various Gram-negative bacteria [23,24,25]. Regarding viral diversity, a metaviromic study conducted on the surface of a smear-ripened cheese revealed the presence of a wide diversity of phage sequence fragments [26]. Host predictions suggested that these phages may target several typical bacteria of smear-ripened cheeses’ community. Recently, the isolation of phages infecting *Brevibacterium aurantiacum* from failed productions of a Canadian smear-ripened cheese was also reported [27]. Together, these results support the need for a deeper exploration of the viral community in smear-ripened cheese ecosystems.

In this study, we extracted viral particles from a French smear-ripened cheese and used it to infect a collection of bacteria isolated from the same cheese. Twelve phages-host combinations were obtained which ultimately allowed for isolating five virulent bacteriophages. Their characterization included morphological characteristics, genome sequencing, host range evaluation and infection capacity at different temperatures. Samples were also collected from the dairy plant producing the studied cheese and analyzed in order to identify their potential reservoirs.

## 2. Material and Methods

### 2.1. Sampling Procedure

Three soft smear-ripened cheeses of the same type, from the same dairy plant, and produced at the same date were purchased in a local food store in November 2019 and directly processed at the lab as triplicates. The rind was gently separated from the core using sterile knives (thickness ∼2–3 mm), mixed using a sterile spatula and used further for microbial counts, bacterial isolation and extraction of viral particles.

### 2.2. Microbiological Analysis 

#### 2.2.1. Enumeration and Isolation of Microorganisms

Bacteria and yeasts were enumerated by plating suitable dilutions (10^−4^ to 10^−7^) of one gram of cheese rind mixed in 9 mL of physiological water (9 g/L NaCl) on three different media as described in [26]. Brain Heart Infusion Agar (BHI, Biokar Diagnostics) supplemented with 50 mg/L amphotericin (Sigma Aldrich, Saint-Louis, MO, USA) was used to count total aerobic bacteria after 48 h of incubation at 28 °C. Man, Rogosa and Sharpe Agar (MRS, Biokar Diagnostics, Allonne, France) supplemented with 50 mg/L amphotericin was used to count lactic acid bacteria after 48 h of incubation at 30 °C under anaerobic conditions. Yeasts were counted on Yeast Extract Glucose Chloramphenicol (YEGC, Biokar Diagnostics, Allonne, France) after 48 h of incubation at 28 °C.

For bacterial isolation, appropriate dilutions (based on the enumeration) were plated on 14 cm diameter Petri dishes containing different growth media. Aerobic bacteria were isolated on rich non-selective BHI medium after incubation at 28 °C for 48 h. Lactic acid bacteria were isolated on MRS medium after incubation at 30 °C for 48 h under anaerobic conditions. Halophilic and halotolerant bacteria were isolated on Marine Agar medium (MA, Difco, BD, Franklin Lakes, NJ, USA). A condition with a final NaCl concentration at 40 g/L (instead of 20 g/L initially) was also tested. MA plates were incubated at three different temperatures (10, 15 and 28 °C) for 48 h to one week. All media were supplemented with 50 mg/L amphotericin to avoid the growth of yeasts and filamentous fungi. For each medium, an initial selection of apparently different morphotypes was performed based on colony morphology (shape and color after 48 h light exposure). A representative of each morphotype was then purified by restriking twice on a new plate. One colony was finally picked and grown in liquid medium (BHI broth, Marine broth or MRS broth) for 24 h before identification and finally stored at −80 °C in glycerol (20% final concentration).

#### 2.2.2. Identification of Bacterial Isolates

For each isolate, genomic DNA extraction was performed as follows: 1–2 mL of an overnight culture in the appropriate broth medium (the same as that used for the isolation step) were centrifuged for 5 min at 5000× *g* and 4 °C. After removing the supernatant, the bacterial pellet was resuspended in 300 µL of TE buffer (10 mM Tris-HCl pH 8.0, 1 mM EDTA, Sigma-Aldrich, Saint-Louis, MO, USA), and 200 mg zirconium beads (BioSpec, Bartlesville, OK, USA), with a 50/50 ratio of 0.1 and 0.5 mm diameters, were added to the tube. Seventy-five µL of the lysozyme-lyticase mix (40 mg/mL and 100 U/mL, respectively, Sigma-Aldrich) were added, and the tube was incubated for 30 min at 37 °C. Forty µL of proteinase K (14 mg/mL, Amresco, VWR, Radnor, PA, USA) and 100 µL of sodium dodecyl sulfate (200 g/L, Sigma-Aldrich) were added, and the tube was incubated for 30 min at 55 °C. After cooling on ice, 500 µL of phenol-chloroform-isoamylic alcohol (25:24:1, pH 8, Sigma-Aldrich, Saint-Louis, MO, USA) were added, and the tube was shaken in a Precellys Evolution homogenizer (Bertin Instruments, Montigny-le-Bretonneux, France) for two 45 s mixing steps at a speed of 9500× *g*. The tube was cooled on ice for 5 min between mixing sequences. After centrifugation at 11,000× *g* for 15 min at room temperature, the aqueous phase was transferred to a Phase Lock Gel tube (Eppendorf, Montesson, France); 500 µL of phenol-chloroform-isoamylic alcohol were added, and the tube was gently mixed. After centrifugation at 11,000× *g* for 5 min at room temperature, the aqueous phase (approximately 700 µL) was transferred to a new Phase Lock Gel tube; 500 µL of chloroform (Sigma-Aldrich, Saint-Louis, MO, USA) were added, and the tube was gently mixed. After centrifugation at 11,000× *g* for 5 min at room temperature, the aqueous phase was recovered in a 2 mL tube, mixed with 2 µL of RNase A (20 mg/mL; Sigma-Aldrich, Saint-Louis, MO, USA) and incubated for 30 min at 37 °C. DNA was precipitated by adding 1200 µL (i.e., twice the aqueous phase volume) of absolute ethanol (Carlo Erba Reagents, Val-de-Reuil, France) and 60 µL (10% of the aqueous phase volume) of sodium acetate (3 M, pH 5.2, Sigma-Aldrich), followed by an incubation period of 30 min at −20 °C. The DNA was recovered by centrifugation at 11,000× *g* for 15 min at 4 °C. The DNA pellet was subsequently washed twice with 1 mL of either 80% or 70% ethanol (*v*/*v*) with a centrifugation step at 11,000× g for 5 min at 4 °C. The pellet was then dried for 30 min at 42 °C and dissolved in 100 to 200 µL of molecular biology grade water.

The small subunit ribosomal RNA gene was amplified using FS1A (5′-AGAGTTTGATCCTGGCTCAG-3′) and FS5H (5′-AAGGAGGTGATCCAGCCGCA-3′) universal primers [28], and the Q5 Hot Start High Fidelity DNA Polymerase (New England BioLabs, Ipswich, MA, USA). Thermal cycling conditions were applied as follows: (i) 5 min at 95 °C for initial denaturation, (ii) 30 cycles of 30 s at 95 °C for denaturation, 30 s at 57 °C for primer annealing, 30 s at 72 °C for elongation, and (iii) 5 min at 72 °C to ensure final elongation. DNA amplicons (expected size of 1500 bp) were assessed on 1.5% *w*/*v* agarose gel and sent for Sanger sequencing to Eurofins Genomics (Köln, Germany). Raw sequences were cleaned under Chromas version 2.6.6, and trimmed sequences were finally compared to the EzBiocloud database using the associated identification tool [29].

### 2.3. Isolation of Bacteriophages, Purification and Concentration

Extraction of the viral fraction from the cheese rind was performed according to protocol P4 detailed in [26] comprising a filtration step and a chloroform treatment. To enhance the chances to isolate phages, an enrichment step was performed as follows. Aerobic bacteria isolated from the cheese rind, including *Brevibacterium aurantiacum*, *Glutamicibacter arilaitensis*, *Psychrobacter aquimaris* and *Psychrobacter cibarius*, were grown overnight in pure cultures in 20 mL of BHI broth supplemented with 10 mM MgSO_4_ and 1 mM CaCl_2_, at 28 °C under agitation at 160 rpm. *Pseudoalteromonas nigrifaciens* was grown in similar conditions, Marine broth (MB) replacing the BHI broth. *Leuconostoc mesenteroides* and *Leuconostoc falkenbergense* were grown in hermetic tubes with 9 mL of MRS broth supplemented with 10 mM MgSO_4_ and 1 mM CaCl_2_, at 28 °C without agitation. One hundred µL of overnight cultures were transferred in fresh medium (20 mL for BHI, 9 mL for MRS) and mixed with 10 µL of the viral fraction to be enriched. The growth conditions were the same except for the temperature of incubation which was lowered to 23 °C for cultures in BHI and MB, as we determined that 28 °C was suboptimal for phage infection in these media.

After centrifugation at 5000× *g* for 10 min at 4 °C, the supernatant was filtrated using 0.22 µm polyethersulfone syringe filters (Sartorius, Göttingen, Germany). Then, 100 µL of the filtrate were used to infect the tested-bacterial isolate through a double-layer spot assay [30]. Double-layer plates were prepared as follows. According to the bacterium tested, the sublayer was made of BHI, MRS or MB containing agar (1.5 %), MgSO_4_ (10 mM) and CaCl_2_ (1 mM). Thirty to one hundred µL of overnight culture of the tested bacterial isolate were added to 5 mL of molten top agarose made of BHI, MRS or Marine broth mixed with 0.3% agarose (MP Biomedicals, Irvine, CA, USA) and supplemented with 10 mM MgSO_4_ and 1 mM CaCl_2_ that were poured on the respective agar plates. The double layer plates were allowed to dry before spotting the enriched phages and were then incubated overnight at 23 °C (28 °C and under anaerobic conditions for MRS). Lysis zones were picked and resuspended in 100 µL of sodium-magnesium (SM) buffer (50 mM Tris ph7.5, 10 mM MgSO_4_, 100 mM NaCl, 1 mM CaCl_2_). Phages were streaked on the same bacterial isolate, and plaques were picked. This step was repeated twice to obtain pure phage stocks. Phage titration was performed using a classical double layer plaque assay [30]. The protocol is close to the spot assay used for phage isolation. One hundred µL of serial dilutions of the phage stock were mixed with the bacterial host and top agarose before pouring on the appropriate agar-medium. After overnight incubation, lysis plaques were enumerated. Phage stocks were stored at 4 °C.

In order to obtain large phage stocks, 100 µL of the pure phages (10^4^ to 10^6^ PFU/mL, depending on phages) were mixed with thirty to one hundred µL of an overnight culture of the propagation host (depending on the strain) and poured on a new double layer plate to obtain confluent lysis. Five ml of SM buffer were poured onto the plates, which were then incubated for one hour at room temperature. The buffer and the top agarose layer were then harvested with a sterile spreader and transferred to a 50 mL tube. After centrifugation for 10 min at 5000× *g* at 4 °C, the supernatant was filtrated. Phage titration was performed as described above. Phage stocks (at least 10^8^ PFU/mL) were stored at 4 °C until use.

### 2.4. Transmission Electronic Microscopy (TEM)

Two milliliters of high titer phage stocks (minimal concentration of 10^8^ PFU/mL) were centrifuged for 1 h at 20,000× *g* and 4 °C. Phage pellets were separately washed and centrifuged for 1 h at 20,000× *g* and 4 °C two times in ammonium acetate (AA) buffer (0.1 M Ammonium Acetate pH 7) before being resuspended in 100 µL of AA buffer. Ten microliters of each phage suspensions were spotted onto a Formwar carbon coated copper grid. Particles were allowed to adsorb to the carbon layer for 5 min, and the excess of liquid was removed. Ten microliters of a staining uranyl acetate solution (1%) were then spotted to the grid for 10 s, and the excess of liquid was removed again. The grid was imaged at 80 kV in a Hitachi HT7700 transmission electron microscope. The dimensions of each phage were determined by averaging the measurements of five separate particles using ImageJ software [31].

### 2.5. Viral Genomic DNA Extraction and Sequencing

Phage stocks at a minimal concentration of 10^8^ PFU/mL were used for DNA extraction following the protocol described in [32] with slight modifications. The DNeasy Blood & Tissue Kit from Qiagen (Hilden, Germany) was replaced by the NucleoSpin Tissue kit from Macherey-Nagel (Hoerdt, France). Final concentrations were measured with a Qubit fluorometer (Thermo Fisher Scientific, Waltham, MA, USA), and the DNA integrity was analyzed on a TapeStation (Agilent Technologies, Santa Clara, CA, USA) using Genomic DNA Screen Tape. Phage DNA was further sequenced using both Illumina and Nanopore technologies.

Regarding Illumina technology, library preparation and sequencing were handled by Eurofins Genomics (Konstanz, Germany). A minimum of 5 million of 150 bp paired-end reads were produced for each phage using a NovaSeq platform (Illumina, San Diego, CA, USA).

For Nanopore sequencing, barcoded genomic DNA sequencing was performed according to the specifications of the Native Barcoding protocol version “NBE_9065_v109_revV_14Aug2019” (Oxford Nanopore Technologies, Oxford, UK). A total of 1–1.5 μg purified DNA from each phage were used to prepare sequencing libraries using the standard Ligation Sequencing Kit SQK-LSK109 (Oxford Nanopore Technologies, Oxford, UK), increasing the DNA repair and end-prep incubation times to 30 min. Sequencing was conducted on MinION Mk1C with FLO-MIN106 (R 9.4.1) flowcells (Oxford Nanopore Technologies, Oxford, UK).

### 2.6. Phage Genome Assembly 

A first draft assembly was constructed from Nanopore reads as described below. The quality of Nanopore reads was evaluated using FastQC (v0.11.8; https://www.bioinformatics.babraham.ac.uk/projects/fastqc/, accessed on 19 April 2022), MultiQC (v1.8) [33] and MinionQC [34]. Porechop (v0.2.3) was used to remove remaining barcodes [35]. After a subsampling of trimmed long reads by Trycycler [36] (v0.4.2; --count 8), genome assembly was performed with three different assemblers, namely Unicycler [37] (v0.4.4; with --long option), Flye [38] (v2.7.1; with --nano-raw--genome-size --plasmids options) and Raven [39] (v1.3.0; default parameters). Then, cleaned contigs of each sample were clustered with Trycycler according to expected size by combining intermediate assemblies of the three tools cited above. We kept the contigs clustered by Trycycler after a manual curation according to the most highly represented clusters and the expected genome size. In accordance with the manual, we then reconciled selected clusters with Trycycler reconcile using cleaned Nanopore reads, aligned sequences with Trycycler msa, partitioned reads with Trycycler partition and finally produced a consensus with Trycycler consensus. Contigs were first polished using trimmed Nanopore reads with Medaka (https://github.com/nanoporetech/medaka, accessed on 19 April 2022) (v1.0.3; -m r941_min_high_g360 option). Then, short Illumina reads were used to polish again the draft assembly with Pilon (v1.23) [40]. For this, reads were first trimmed with Trimmomatic v0.39 [41] (options: ILLUMINACLIP:TruSeq3-PE.fa:2:30:10, LEADING:3, TRAILING:3, SLIDINGWINDOW:4:20, MINLEN:125). The final quality of the assembled contigs was assessed with QUAST v5.0.2 [42].

A second-draft assembly was produced from trimmed Illumina reads only and assembled with SPAdes (v3.13.1) with the --only-assembler option and increasing kmer values -k 21,33,55,77,99,127 [43]. PhageTerm [44] was used to predict the genomic termini and phage packaging strategy using both trimmed Illumina reads and corresponding assembled contigs. Automatically rearranged contigs were thus obtained and aligned with the contigs originating from the first draft assembly. When necessary, this second assembly was used to correct the first assembly manually and produce the final phage genomes.

### 2.7. Structural and Functional Annotation 

For each polished genome, open reading frames (ORFs) were predicted with the RAST server [45] with the following parameters: Domain = Viruses, Genetic code = 11, RAST annotation scheme = RASTtk. Afterwards, each ORF was manually annotated using a combination of the following tools: (i) HHpred against the PDB-mmCIF70_12_Oct database [46], (ii) Blast [47] against the nr/nt database from the National Center for Biotechnology Information (NCBI)) and the Conserved Domains Database (CDD) [48,49], (iii) PHROGs [50] and (iv) Virfam [51]. The presence of virulence genes was searched by comparing all the coding DNA sequences (CDS) to the Virulence Factor DataBase [52] by BlastX (thresholds: 90% identity, 60% coverage). The presence of antibiotic resistance genes were searched using ResFinder [53] using the same thresholds.

### 2.8. Genome-Based Classification of Phages

Assembled genomes were compared to the nr/nt NCBI database using BLASTn (https://blast.ncbi.nlm.nih.gov/Blast.cgi, accessed on 21 April 2022), with and without specifying the “Viruses” option in the search. According to the International Committee on Taxonomy of Viruses (ICTV), two phages were considered to respectively belong to the same species or genus if the genome were reciprocally more than 95% or 70% identical at the nucleotide level over their (almost) full genome length [54]. These taxonomic values were calculated using BLASTn multiplying % identity by % coverage, which is one of the tools proposed in [54]. In order to evaluate putative higher phage taxonomic affiliation (subfamily or family ranks), we used the ViPTree web server [55] (https://www.genome.jp/viptree, accessed on 22 April 2022) which is a proteome-based clustering tool that generates “proteomic trees” of phage genome sequences based on genome-wide similarities computed by tBLASTx. Finally, comparative genomics were performed between the newly sequenced phages and their closest relative(s) with Easyfig (v2.2.5) [56]. The shed of red lines precisely connects regions of adjacent phages that have tBLASTx identity from 30% to 100% over at least 100 or 150 bp depending on phage genomes.

### 2.9. Host Range Determination

To assess the host range of the newly isolated phages, we used a set of bacterial strains and isolates listed in Table 1. Briefly, we tested the sensitivity of each phage against (i) up to 13 isolates from the washed-rind cheese (see Section 2.2.2) belonging to the same bacterial species as the propagation strain where possible (randomly selected), (ii) collection strains isolated form other dairy products or different environments and (iii) collection strains belonging to the same genus but different species as the propagation strain. Bacterial sensitivity was assessed through a spot assay experiment, as described in [57].

### 2.10. Effect of Temperature on Phage Infection

Serial dilutions for each phage (from pure stocks to 10^−8^) were performed in SM Buffer and spotted (5 µL) on five double-agar plates containing a pure culture of the appropriate recipient strain (*G. arilaitensis* G65 for Volaire, *G. arilaitensis* G51 for Montesquieu, *B. aurantiacum* B67 for Rousseau, *L. falkenbergense* 91 for Diderot or *Psychrobacter aquimaris* 87 for D’Alembert). Plates were incubated for 24 to 48 h either at 12 °C, 16 °C, 20 °C, the original temperature of isolation (25 °C for Voltaire, Montesquieu, Rousseau and D’Alembert; 28 °C for Diderot) or 30 °C. For each temperature, the phage titer was determined by counting lysis plaques at the lowest possible dilution and then compared to that obtained at the phage isolation temperature (25 °C or 28 °C).

### 2.11. Identification of Phage Reservoirs in a Dairy Plant

Five sample types were collected in the production unit producing the studied surface ripened cheese: milk after inoculation with acidification ferments, salting tables (after cleaning), cheese turning line (after cleaning), and two washing solutions. For each sample type, three replicates were collected at weekly intervals during May 2021, that is, one-and-a-half years after the first viral extractions from the cheese rind used for the isolation of phages.

A virome of each sample was prepared using a procedure adapted to each sample type. Liquid samples (milk and the two washing solutions, 50 mL each) were centrifuged at 300× *g* for 10 min at 4 °C, and the supernatant was filtrated on 0.22 µm polyethersulfone syringe filters (Sartorius). Wipes from the solid surfaces (salting tables and cheese turning lines) were placed in flasks containing 100 mL of SM buffer and agitated overnight at 4 °C. They were then manually and aseptically wrung, and the liquid was filtered on 0.22 µm polyethersulfone syringe filters (Sartorius). After filtration, phage precipitations were performed by adding 10% (*w*/*v*) polyethylene glycol 8000 (Sigma Aldrich) and 1 M NaCl. After an overnight incubation at 4 °C, precipitates were centrifuged at 6000× *g* for 60 min at 4 °C. The pellet was resuspended in 2 mL of SM buffer and stored at 4 °C.

To search for phages in the collected samples from the dairy plant, spot assays were performed. For this, we selected as recipients the bacterial hosts sensitive to the five phages isolated from the cheese rind. Briefly, 10 µL of the viral extract from the production sample were spotted on a double-agar plate containing a pure culture of either *G. arilaitensis* G65, *G. arilaitensis* G51, *B. aurantiacum* B67, *L. falkenbergense* 91 or *Psychrobacter aquimaris* 87 bacterial isolates (Table 1). In cases where no lysis was detected by direct plating, a phage enrichment was performed by adding 10 µL of the viral extract from the production sample to the tested strain in 20 mL of the appropriate broth medium and incubation overnight at 23 °C. The culture supernatant was then filtered and used for the spot-testing.

When present, plaques were picked with a sterile loop and resuspended in 30 µL of SM buffer. Then, PCR amplification using diagnostic primers for each of our 5 characterized phages (Table 2) and Sanger sequencing (Eurofins Genomics) was conducted to test whether the phage detected in the sample corresponded to the one isolated from the cheese rind. Thermal cycling conditions were applied as follows: (i) 5 min at 95 °C for initial denaturation, (ii) 30 cycles of 30 s at 95 °C for denaturation, 30 s at the appropriate annealing temperature, 30 s at 72 °C for elongation and (iii) 5 min at 72 °C to ensure final elongation. DNA amplicons were assessed on 1.5% *w*/*v* agarose gel and sent for Sanger sequencing to Eurofins Genomics (Köln, Germany). Raw sequence reads were cleaned under Chromas version 2.6.6 and aligned to the targeted sequences from isolated phage genomes.

## 3. Results

### 3.1. Construction of a Bacterial Collection from the Cheese Surface

Enumeration of total viable counts on BHI, lactic acid bacteria on MRS and yeasts on YEGC gave 5.1 × 10^9^ CFU/g, 1.7 × 10^7^ CFU/g and 2.2 × 10^8^ CFU/g of the cheese rind, respectively, was conducted. From the three sampled cheeses, 203 bacterial strains were isolated on four different media and selected on the basis of colony morphology in order to maximize the final diversity present in the collection. After purification, bacteria were identified according to the full-length sequence of the SSU rRNA gene. Most isolates belonged to six species, i.e., *Glutamicibacter arilaitensis* (78 isolates), *Psychrobacter cibarius* (46), *Psychrobacter aquimaris* (25), *Leuconostoc falkengergense* (19), *Leuconostoc mesenteroides* (10) and *Halomonas nigrificans* (9). Remaining bacterial isolates corresponded to sub-dominant species such as *Brevibacterium aurantiacum* (2), *Staphylococcus equorum* and *Vibrio hibernica*. This collection of bacterial isolates was used to isolate bacteriophages from the same samples.

### 3.2. Five Bacteriophages Isolated from the Cheese Surface

We explored the presence of bacteriophages in the rind of the same cheese used for bacterial sampling by screening the bacterial collection using a classical double-layer spot assay (see Methods).

For three of the six most frequently isolated species, no phages were isolated, namely *Psychrobacter cibarius*, *Leuconostoc mesenteroides* and *Halomonas nigrificans*. Four virulent bacteriophages could be isolated, however, from the three other dominating species: two *Glutamicibacter arilaitensis* infecting phages, Voltaire (infecting isolate G65) and Montesquieu (infecting isolate G51), one *Leuconostoc falkenbergense* infecting phage, Diderot (infecting isolate 91) and one *Psychrobacter aquimaris* infecting phage, D’Alembert (infecting isolate 87). Interestingly, we could also isolate a phage infecting the sub-dominant species *Brevibacterium aurantiacum*, Rousseau (on isolate B67).

#### 3.2.1. All Cultivable Phages Are Tailed Phages

Electron micrographs of these phages showed that they were all tailed and therefore belonged to the *Caudoviricetes* class (Figure 1). Their main morphological characteristics are summarized in Table 3. D’Alembert is a myophage (long, contractile tail); Voltaire is a podophage (i.e., it has a short tail), while Rousseau, Diderot and Montesquieu are siphophages (long non-contractile tail).

#### 3.2.2. All Five Phages Have Narrow Host Ranges

Each phage host spectrum was evaluated by spot testing using three distinct groups of strains or isolates, namely: (i) isolates of the same species as the propagation strain obtained from the same cheese, (ii) collection strains belonging to the same species as the propagation strain but obtained from other sources (other dairy products or various environments) and (iii) collection strains belonging to the same genus but different species as the propagation strain (Table 4).

The five newly isolated phages all had a narrow host range, being able to infect only one to seven isolates of a single species and all originating from the studied cheese. Interestingly, the infection profiles of Voltaire and Montesquieu phages, both infecting *G. arilaitensis*, were completely different (Appendix A).

#### 3.2.3. Effect of Temperature on Phage Infection

Phage infections were performed at different incubation temperatures ranging from 12 °C to 30 °C (Appendix A). The infection success of *Psychrobacter* phage D’Alembert and *Glutamicibacter* phage Montesquieu was barely constant from 12 to 25 °C but dropped at 30 °C (almost 2 and 4 logs for D’Alembert and Montesquieu, respectively). A moderate reduction in the titer was observed for *Glutamicibacter* phage Voltaire (almost 1 log) at 25 and 30 °C compared to lower temperatures, and, on the contrary, at 12 °C for *Leuconostoc* phage Diderot compared to higher temperatures. *Brevibacterium* phage Rousseau infection was not affected by the temperature within the tested range.

#### 3.2.4. Uncovering Three Completely New Phage Genomes

The genomes of the five phages isolated from the cheese rind were sequenced using both Nanopore and Illumina technologies. The main sequencing and assembly information are summarized in Table 5. Briefly, the number of raw Nanopore reads obtained was comprised between 15 k and 1.7 M, and the number of raw Illumina reads between 5.5 M and 8.9 M. As expected, Nanopore sequencing produced long reads, from 3074 to 9872 bases on average, which facilitated complete genome assembly.

Genome sizes ranged from 18 kb for *Glutamicibacter* phage Voltaire to 92 kb for *Psychrobacter* phage D’Alembert. Gene density was high (1.6 genes/kb in average), as is usual for phage genomes, although we noticed that Montesquieu had a slightly lower gene density (1.3 genes/kb). For *Glutamicibacter* phage Montesquieu, the assembly of Nanopore reads did not produce a unique contig.

The completeness of the sequenced genomes was further investigated (Table 6). Voltaire was complete as indicated by the presence of 176 bp inverted terminal repeats, and D’Alembert as well, with 5.7 kb-long direct terminal repeats (DTRs). Interestingly, D’Alembert DTRs were found thanks to PhageTerm, whereas, in the single contig generated upon the SPAdes assembly from Illumina reads, one of the repeats was missing. Regarding Rousseau and Montesquieu, Illumina assembly displayed a single contig with 127 bp artefactual DTRs, indicating completeness as well. Finally, Diderot was not assembled into a single contig from Illumina reads, but its assembly from Nanopore reads produced a single 27.1 kb contig, highly similar to a known phage over its total length, suggesting genome completeness (see below). Overall, both assemblies gave complementary information and allowed for conclusions on the completeness of 4 out of 5 phage genomes and suggested completion for the last one. With respect to encapsidation modes, PhageTerm predicted Rousseau and Diderot as cos phages with 3′ extensions, Montesquieu as a pac phage, and D’Alembert uses long DTRs.

Each genome was further characterized by performing manual annotation of each ORF. We used the ViPTree webserver to perform comparisons of global protein content of our phages with those available as of March 2022 in this interface (Figure 2 and Figure 3). Then, each phage genome was aligned to one or two close relatives with Easyfig (Figure 4). A table gathering the eleven auxiliary metabolic genes (AMGs) detected can be found in Appendix A. Here, AMGs are encoded by all phages but Voltaire. Finally, no virulence genes nor antibiotic resistance genes were detected on the five newly assembled genomes.

*Glutamicibacter* phage Voltaire encodes 26 ORFs, of which 14 had function predictions. It has no AMGs but possesses three genes coding for enzymes involved in lysis: an amidase (VOLT_14), an endopeptidase (VOLT_19) and a holin (VOLT_21). It represents the first completely new phage genome uncovered by the study. According to ViPTree, the closest phages to Voltaire are Mendel and Anjali, two small and closely related podophages infecting *Arthrobacter* (Figure 2A). Voltaire is more distantly related to *Actinomyces* podophage Av-1, sharing weak homologies with the polymerase B (VOLT_6), major capsid protein (VOLT_9) and major tail protein (MTP, VOLT_15) (Figure 4A). The ViPTree positioning leads to proposing that *Glutamicibacter* phage Voltaire may represent a new genus within the *Salasmaviridae* family.

*Glutamicibacter* phage Montesquieu comprises 62 ORFs of which 36 have function predictions. It possesses two predicted AMGs, an ABC transporter (MONT_40) and an aminocyclopropane-1-carboxylate deaminase (MONT_41). No significant BLASTn hits were obtained by comparing the Montesquieu sequence to the NCBI database (nr and viruses), making this latter the second new phage characterized in the study. According to the ViPTree analysis, the closest phages to Montesquieu are a clade of siphophages infecting *Brevibacterium* or *Arthrobacter*, from which Montesquieu probably represents a new genus (Figure 2B). The genomic comparison of Montesquieu with its two closest ViPTree neighbors, *Arthrobacter* phage TripleJ and *Brevibacterium* phage LuckyBarnes showed a similar genetic organization but little sequence identity at the protein level, mostly within the structural module (Figure 4B). Montesquieu owns proteins related to the tail but no sheath protein, which is consistent with the siphophage morphotype observed in TEM.

*Brevibacterium* phage Rousseau has 71 ORFs of which 41 have functional predictions. It possesses three AMGs: a putative glutaminyl cyclase (ROUS_25), a thioredoxin-like protein (ROUS_48) and an S-adenosylmethionine-dependent methyltransferase (ROUS_51). With no homologues in the NCBI database, Rousseau is the third completely new phage uncovered by the study and represents a new genus. According to ViPTree, Rousseau is only very distantly related to phages infecting *Propionibacterium* (Figure 3A). A comparison of the genetic maps of Rousseau and AGM1, a phage previously isolated from a smear-ripened cheese wash solution, shows similar organizations despite their lack of genetic relatedness (Figure 4C). The Rousseau tail module supports its siphophage morphological characteristics observed in TEM.

*Psychrobacter* phage D’Alembert comprises 158 ORFs of which 56 have functional predictions. It has several AMGs, namely: a nucleoside triphosphate pyrophosphohydrolase (DAL_25), an Ntn_hydrolase-like protein (DAL_45), an S-adenosylmethionine-dependent methyltransferase (DAL_53), a putative chaperonin (DAL_56), an endolytic peptidoglycan transglycosylase (DAL_57), a putative antitoxin (DAL_112) and a thioredoxin glutathione reductase (DAL_125). It represents a new phage genus, sharing a third of its genome with *Vibrio* phage vB_VhaM_VH-8 (84% nt identity). According to ViPTree, both phages, D’Alembert and VH-8, are distantly related to myophages infecting *Acinetobacter* (Figure 3C). Genomic comparison confirmed that D’Alembert is closer to VH-8 than to *Acinetobacter* phage vB_AbaM_Acibel004, with which it showed only little gene synteny and sequence homology for a few proteins (Figure 4E). The fact that a sheath protein is encoded on the D’Alembert genome supports the direct myophage morphotype observation performed using TEM.

*Leuconostoc* phage Diderot has 40 ORFs of which 29 have a functional prediction. It shares a very strong nucleotidic identity with *Leuconostoc* siphophage LN03 (98% identity and 98% coverage for both genomes). Both Diderot and LN03 harbor an AMG coding for a ribonuclease Z (DID_7 and LN03_7). A remarkable difference among the genomes is located on LN03_2 and DID_2: although both proteins are predicted endodeoxyribonucleases, they share less than 30% protein identity (Figure 4D), suggesting a recent exchange. Based on ICTV taxonomic criteria, Diderot belongs to the same species as LN03, the *Limdunavirus* genus and the *Mccleskeyvirinae* subfamily, which is confirmed by ViPTree analysis (Figure 3B).

#### 3.2.5. Most of the Identified Phages Are Also Present in the Dairy Plant

In order to identify potential reservoirs of the five phages studied, the dairy plant—producing the cheese from which the phages were isolated—was investigated. The viral fractions resulting from each sample were first tested on five indicator strains with no enrichment step (each strain is sensitive to one of the five isolated phages described above) through a spot assay. If no plaques were observed, a second spot assay on the same strains was performed after enrichment (see Methods). The results are summarized in Figure 5.

Samples obtained from the two washing solutions did not produce lysis plaques using any of the tested strains. However, confluent lysis or clear lysis plaques were obtained using samples from milk, salting tables and the cheese turning line. More precisely, the cheese turning line represented the main reservoir for virulent phages infecting bacteria growing on the rind of this cheese. Indeed, confluent lysis spots were detected for four of the five recipient strains used in the assay (all but *G. arilaitensis* G65, the propagation strain of phage Voltaire) after infection with these samples. Furthermore, this result was observed three times at a one-week interval, revealing the persistence of the corresponding phages’ species on the cheese turning line despite several cleaning cycles.

Salting tables represented a second, more restricted but persistent reservoir for dairy phages infecting *L. falkenbergense*. Lysis plaques (rather than confluent lysis zones) were repeatedly observed with *L. falkenbergense* 91 as the tested strain. This strain was also sensitive to phages coming from one of the three milk samples, indicating that inoculated milk can occasionally contain virulent phages.

Overall, the second spot assay (after enrichment) allowed for the observation of plaques for the same samples as the first one. In order to determine if the phages detected in the potential reservoirs through this experiment were indeed related to the ones previously isolated from the cheese surface, specific primers were designed for each phage (Table 2) and used to amplify and sequence the genetic material present in the different lysis zones or plaques. The sequences amplified from lysis zones detected using *G. arilaitensis* G51, *B. aurantiacum* B67 and *P. aquimaris* 87 indicator strains presented high nucleotidic identity (from 98.4% to 100%) with phages Montesquieu, Rousseau and D’Alembert, respectively (Figure 5). This suggested that these phages, or their close relatives, were still present and infectious in the dairy plant one-and-a-half years after isolation from the cheese surface. Regarding the lysis plaques obtained with *L. falkenbergense* 91, a PCR product of the size expected for Diderot was also observed, but, depending on the sample, its sequence was not exactly identical to Diderot (ranging from 95.78 to 99.1%) (Figure 5). This may indicate that multiple phages capable of infecting this strain co-exist or evolve in the dairy plant.

## 4. Discussion

In this study, we first isolated and identified a collection of bacterial isolates from the rind of a French smear-ripened cheese to use it in a second phase for the isolation of phages from the same cheese. The biodiversity of the bacteria isolated during this research, comprising six main species, is typical of this kind of cheese. Indeed, the bacterial community of the surface of washed-rind cheeses generally comprises several distinct groups such as non-starter lactic acid bacteria (e.g., *Leuconostoc* spp.), staphylococci (e.g., *S. xylosus*, *S. equorum*), coryneform bacteria (e.g., *Glutamicibacter arilaitensis*, *Brevibacterium aurantiacum*, *Corynebacterium variabile*, *Microbacterium gubbeenense*) and Gram-negative bacteria (e.g., *Alcaligenes faecalis*, *Halomonas* spp., *Psychrobacter* spp., *Hafnia alvei*, *Proteus* spp., *Vibrio* spp., *Pseudoalteromonas* spp.) [24,58,59,60].

In contrast, only two isolates of *Brevibacterium aurantiacum*, which is generally added as a ripening culture in washed-rind cheeses produced worldwide [61], were obtained, indicating its low ability to outcompete the resident microbiota in this particular cheese. This trend was already observed for several commercial smear starter strains [62,63] and is widely discussed in the literature [64,65], although the reasons explaining their lack of fitness is not fully understood yet. One reason may be the presence of phages infecting such species in cheese. Indeed, one was successfully isolated in this study from a French smear-ripened cheese (*Brevibacterium* phage Rousseau), and a collection of sixteen phages (represented by *Brevibacterium* phage AGM1) was also recently isolated from similar Canadian products or their production environment [27].

Unlike *B. aurantiacum*, *Glutamicibacter arilaitensis* (formerly *Arthrobacter arialitensis*) represented the most frequently isolated species in the bacterial collection established from the studied cheese. This yellow-pigmented bacteria is one of the major bacterial species found at the surface of smear-ripened cheeses [66,67]. It can be either deliberately inoculated as a ripening culture or naturally present in cheese, and previous work indicated the possible co-existence of multiple strains of *G. arilaitensis* in a single cheese product [62]. Interestingly, two genetically different phages (Voltaire and Montesquieu) with non-overlapping host ranges were isolated in this study from the same cheese rind. Whether the observed phage sensitivity could be related to the co-existence of distinct strains of *G. arilaitensis* within the studied cheese remains to be elucidated. Indeed, phages have already been identified as a key biotic factor favoring the maintenance of intra-species diversity in undefined starter cultures [68,69]. Another explanation would be that two populations of the same strain are present, differing only by their phage resistance profiles in terms of CRISPR diversity as suggested in [70].

Four out of the five newly described phages, namely *Glutamicibacter* phage Voltaire, *Glutamicibacter* phage Montesquieu, *Brevibacterium* phage Rousseau and *Psychrobacter* phage d’Alembert, shared only little sequence homology with previously sequenced phages. With the increasing number of phage genomes available, genome-based taxonomy is now used for phage classification. Specific requirements, including sequence identity thresholds for species and genus levels, have been proposed for rank-based demarcation of tailed phages [54]. Based on these criteria, the four above-mentioned phages would represent at least four new genera. This result illustrates the under-representation of phages infecting cheese-rind bacteria in public databases. Therefore, the cheese rind should be considered as an attractive environment for the discovery of new phages with potential interest for the cheese industry and ferment producers.

AMGs were found in four of the five studied phages. These genes encode proteins similar to those used in the host metabolism and are supposed to boost metabolic steps that might be bottlenecks in the phage reproduction process [71]. They are mainly implied in the protein and nucleic acid metabolism. Among them, two were, to our knowledge, never described within phage genomes until now. Phage Rousseau ROUS_25 protein is a putative glutaminyl cyclase, named QC for short (HHpred likelyhood probability 99.19% to PDB QC 3NOL). Glutaminyl cyclases are well characterized in eukaryotic organisms. This enzyme catalyzes the cyclization of N-terminal glutamine residues to the pyroglutamate of various proteins [72]. QC were more recently found as well in bacteria, and their function appears essential for *Porphyromonas gingivalis* growth [73,74]. The QC function in phage Rousseau remains to be established; it may help virion proteins to be more resistant against host proteolytic activity. Interestingly, a distant homolog of this protein is encoded in the genome of several *Brevibacterium* species (45% amino-acid identity). It is also similar to a *Brevibacterium iodinum* phage gene (Lucky Barnes, accession YP_009792202.1), which is annotated as “minor tail protein”. The second new AMG is present in *Glutamicibacter* phage Montesquieu, and encodes an aminocyclopropane-1-carboxylate deaminase (ACCD, MONT_41), just downstream from an ABC transporter. This cassette may contribute to improved amino-acid import and/or synthesis. One AMG present in the D’Alembert genome encoded a putative antitoxin (DAL_112) and might be considered as a host takeover function. Indeed, it is related to a Staphylococcus aureus antitoxin (HHpred likelihood probability 98.56% to PDB 6L8G), and antitoxins encoded by phages can protect themselves against host-produced toxins [75,76].

The host range of the five newly isolated phages is narrow and limited to a few sensitive isolates, which were exclusively obtained from the same tested cheese. All tested collection strains (not originating from that particular cheese), even the ones belonging to the same species as the indicator strain, were resistant. Similar results were reported for *Propionibacterium freudenreichii* phages isolated from Swiss hard cheese [18], and it is assumed that most phages possess a narrow host range [77]. Thus, with the aim of isolating bacteriophages from a given food sample, one should privilege building a specific collection of bacterial isolates from the same sample (sharing the same ecological niche), and, then, use it to search for phages through spot assays with or without enrichment. This approach may, however, favor the isolation of virulents at the expense of temperate phages due to superinfection immunity. Indeed, temperate phages originate from, or generate, bacterial lysogens, which, when isolated in the same environment and used as indicators, will prevent phage growth and plaque detection [78].

Looking for the origin of the isolated phages, we investigated five different types of samples collected in the cheesemaking plant producing the studied washed-rind cheese as potential reservoirs. Four phages (Rousseau, Montesquieu, Diderot and D’Alembert) were directly detected on the cheese turning line and one on the salting tables (Diderot). For Diderot, PCR-sequencing results revealed some nucleotide variance suggesting the co-occurrence of several closely related phages within the dairy plant, but further experiments are required to confirm this observation. The fact that the non-enriched viral fraction allows for the observation of plaques as well as enriched ones indicates that the level of contamination is non-negligible. Interestingly, in such samples, the positive detection of the different phages was observed three times at weekly intervals, indicating the persistence of these phages on working surfaces of the cheese plant despite regular cleaning procedures. Furthermore, the samples from the dairy plant were obtained almost 18 months after the isolation of phages from the cheese surface. This result indicates a long-term persistence of the four phages within the production environment and especially on manufacturing surfaces. On the other hand, milk after inoculation or the washing solutions should not be considered as major reservoirs since no targeted phages were repeatedly detected in such samples. Phage contamination in dairy plants has already been observed for a long time, but most studies were focused on phages infecting LAB starters. According to the literature, the most probable sources of dairy phages are the starter cultures themselves, as some strains carry prophages that can evolve towards virulence later during the process [79], milk, whey, airborne particles and processing surfaces [15,19,80,81,82]. Previous studies conducted on cheesemaking facilities producing Gubbeen [83], or fresh, bloomy-rind and washed-rind cheeses [2], revealed that dominant bacterial and fungal taxa present on cheese, and mainly the non-inoculated ones, are also contaminating processing surfaces. Our study therefore suggests that the same applies for four bacteriophages infecting the rind bacteria.

## 5. Conclusions

Virulent bacteriophages infecting four of the main bacterial species living on the rind of a smear-ripened cheese were isolated and characterized. This provides the formal evidence that a diverse viral community co-occurs in this ecosystem along with the well-described bacterial and fungal communities, as previously suggested by viral metagenomics data. The low genomic relatedness of most of the newly isolated phages with currently known phages underlines the lack of knowledge regarding the viral fraction of the cheese ecosystem. Microbial communities of the cheese surface being largely involved in the typical sensory attributes and quality of the final products, further understanding about the role of such entities on cheese microbial ecology and finally their impact on cheese ripening would now be desirable.

## Figures and Tables

**Figure 1 viruses-14-01620-f001:**
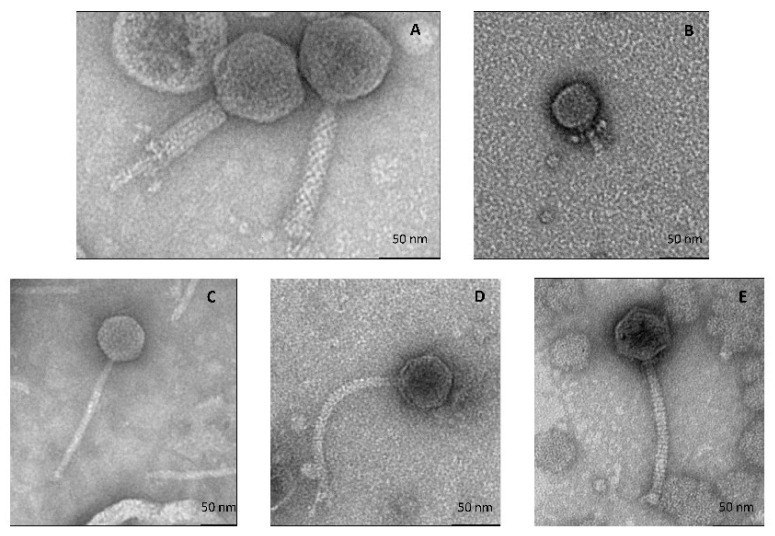
Transmission electron micrographs of 5 phages isolated from cheese rind. (**A**), *Psychrobacter* phage D’Alembert (contracted form at left), (**B**) *Glutamicibacter* phage Voltaire, (**C**) *Glutamicibacter* phage Montesquieu, (**D**) *Brevibacterium* phage Rousseau and (**E**) *Leuconostoc* phage Diderot.

**Figure 2 viruses-14-01620-f002:**
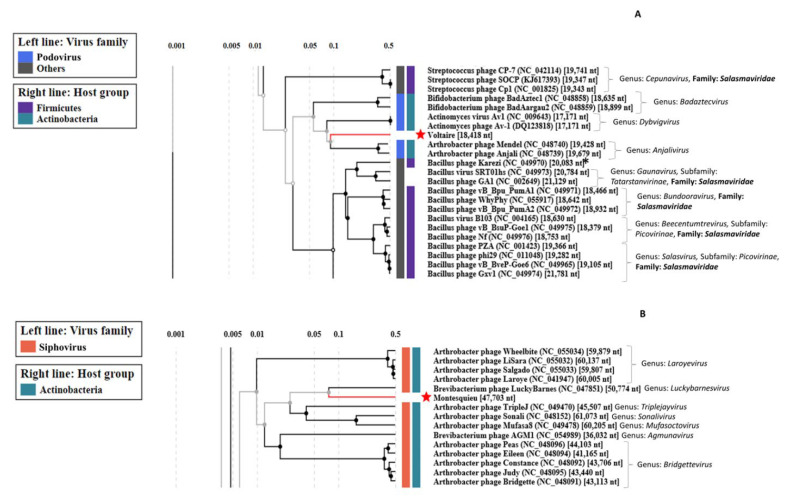
Screenshots of proteomic trees of (**A**) Voltaire and (**B**) Montesquieu and related phages, computed with ViPTree. *: Genus: Karezivirus, Subfamily: Tatarstanvirinae, Family: Salasmaviridae. Red stars indicate the position of the newly sequenced phage genomes.

**Figure 3 viruses-14-01620-f003:**
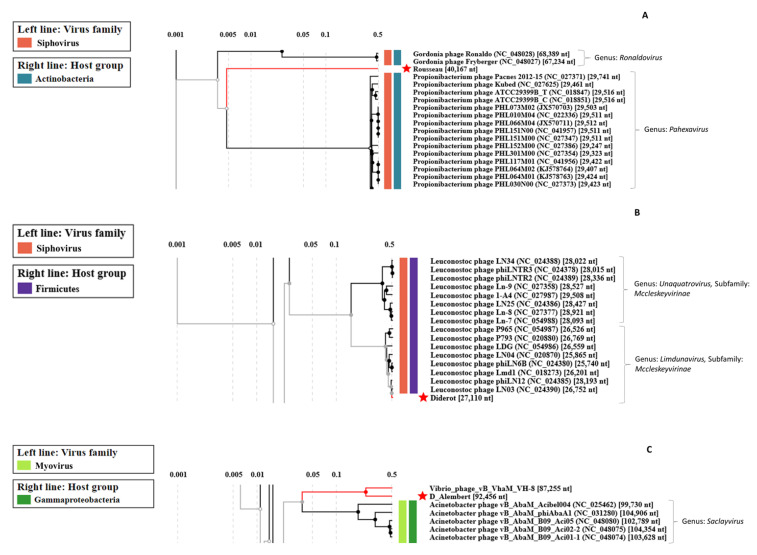
Screenshots of proteomic trees of (**A**) Rousseau, (**B**) Diderot, (**C**) D’Alembert and related phages, computed with ViPTree. Red stars indicate the position of the newly sequenced phage genomes.

**Figure 4 viruses-14-01620-f004:**
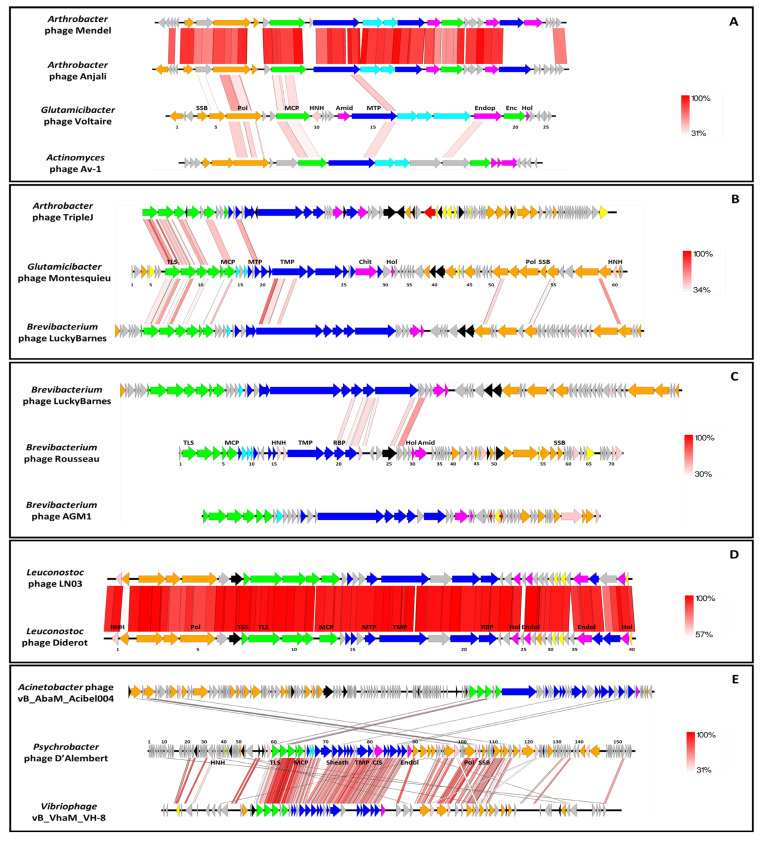
Schematic representation of the phage genomes and comparisons to their closest relatives. (**A**) *Glutamicibacter* phage Voltaire. (**B**) *Glutamicibacter* phage Montesquieu. (**C**) *Brevibacterium* phage Rousseau. (**D**) *Leuconostoc* phage Diderot. (**E**) *Psychrobacter* phage D’Alembert. Each line represents a phage genome, and each arrow represents an ORF. Red shade lines and percentages indicate tBLASTx identity between two genes. A minimum BLAST hit length of 100 nt (150 for d’Alembert) and with at least 30% tBLASTx identity were set. Gene functions are color-coded and detailed (yellow: transcriptional regulation, orange: DNA metabolism, green: DNA packaging and head, light blue: head to tail, dark blue: tail, pink: HNH endonuclease, fuchsia: lysis, black: auxiliary metabolic genes, grey: hypothetical proteins). List of abbreviations: Amid = amidase; Chit = chitinase; CIS = Contractile Injection System; Enc = encapsidation protein; Endop = endopeptidase; Endol = endolysin; HNH = HNH homing endonuclease; Hol = holin; MCP = Major Capsid Protein; MTP = Major Tail Protein; Pol = polymerase; RBP = Receptor-Binding Protein; SSB = Single-Strand Binding protein; TLS = Terminase Large Subunit; TMP = Tail tape Measure Protein; TSS = Terminase Small Subunit.

**Figure 5 viruses-14-01620-f005:**
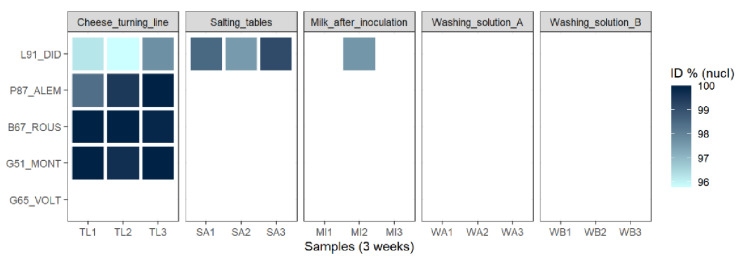
Sensitivity of indicator strains to phages present in different samples collected within the dairy plant.

**Table 1 viruses-14-01620-t001:** Bacterial hosts tested for their sensitivity to isolated phages.

Strain or Isolate	Isolation Source	Tested Phages
*Glutamicibacter arilaitensis* G16	Studied cheese	Voltaire and Montesquieu
*Glutamicibacter arilaitensis* G26	Studied cheese	
*Glutamicibacter arilaitensis* G33	Studied cheese	
*Glutamicibacter arilaitensis* G43	Studied cheese	
*Glutamicibacter arilaitensis* G51	Studied cheese	
*Glutamicibacter arilaitensis* G52	Studied cheese	
*Glutamicibacter arilaitensis* G53	Studied cheese	
*Glutamicibacter arilaitensis* G65	Studied cheese	
*Glutamicibacter arilaitensis* G119	Studied cheese	
*Glutamicibacter arilaitensis* G135	Studied cheese	
*Glutamicibacter arilaitensis* G183	Studied cheese	
*Glutamicibacter arilaitensis* G186	Studied cheese	
*Glutamicibacter arilaitensis* G201	Studied cheese	
*Glutamicibacter arilaitensis* DSM 16368	Reblochon cheese	
*Glutamicibacter bergerei* DSM 16367	Camembert cheese	
*Glutamicibacter nicotianae* DSM 20123	Air of tobacco warehouses	
*Glutamicibacter uratoxydans* DSM 20647	Humus soil	
*Brevibacterium aurantiacum* B20	Studied cheese	Rousseau
*Brevibacterium aurantiacum* B67	Studied cheese	
*Brevibacterium aurantiacum* 2M23	Cheese	
*Brevibacterium aurantiacum* FME9	Cheese	
*Brevibacterium aurantiacum* FME34	Cheese	
*Brevibacterium aurantiacum* FME43	Cheese	
*Brevibacterium aurantiacum* FME45	Cheese	
*Brevibacterium aurantiacum* FME48	Cheese	
*Brevibacterium aurantiacum* FME49	Cheese	
*Brevibacterium aurantiacum* ATCC 9174	Cheese	
*Brevibacterium aurantiacum* ATCC 9175 (DSM 20426)	Camembert cheese	
*Brevibacterium aurantiacum* 25	Camembert cheese	
*Brevibacterium aurantiacum* 299	Camembert cheese	
*Brevibacterium aurantiacum* B3	Langres cheese	
*Brevibacterium aurantiacum* CAM-4	Camembert cheese	
*Brevibacterium aurantiacum* CAM 12C	Camembert cheese	
*Brevibacterium casei* CIP 102111 (DSM 20657)	Cheese	
*Brevibacterium epidermidis* NCDO 2286T (DSM 20660)	Skin	
*Brevibacterium iodinum* ATCC 49514T (DSM 20626)	Skin	
*Brevibacterium linens* ATCC 9172 (DSM 20425)	Cheese	
*Brevibacterium sandarakinum* DSM 22082	Wall surface	
*Leuconostoc falkenbergense* 90	Studied cheese	Diderot
*Leuconostoc falkenbergense* 91	Studied cheese	
*Leuconostoc falkenbergense* 92	Studied cheese	
*Leuconostoc falkenbergense* 93	Studied cheese	
*Leuconostoc falkenbergense* 96	Studied cheese	
*Leuconostoc falkenbergense* 98	Studied cheese	
*Leuconostoc falkenbergense* 99	Studied cheese	
*Leuconostoc falkenbergense* 114	Studied cheese	
*Leuconostoc falkenbergense* 116	Studied cheese	
*Leuconostoc mesenteroides* 88	Studied cheese	
*Leuconostoc mesenteroides* 89	Studied cheese	
*Leuconostoc mesenteroides* 95	Studied cheese	
*Leuconostoc mesenteroides* 97	Studied cheese	
*Leuconostoc mesenteroides* 101	Studied cheese	
*Leuconostoc mesenteroides* 102	Studied cheese	
*Leuconostoc mesenteroides* 107	Studied cheese	
*Leuconostoc mesenteroides* 108	Studied cheese	
*Leuconostoc mesenteroides* 113	Studied cheese	
*Leuconostoc mesenteroides* 115	Studied cheese	
*Leuconostoc citreum* MSE2	Milk	
*Leuconostoc lactis* NCW1	Cheese	
*Leuconostoc mesenteroides* ssp. *cremoris* DSM 20346	Cheese	
*Leuconostoc pseudomesenteroides* MSE7	Cheese	
*Psychrobacter aquimaris* 15	Studied cheese	D’Alembert
*Psychrobacter aquimaris* 54	Studied cheese	
*Psychrobacter aquimaris* 59	Studied cheese	
*Psychrobacter aquimaris* 60	Studied cheese	
*Psychrobacter aquimaris* 69	Studied cheese	
*Psychrobacter aquimaris* 87	Studied cheese	
*Psychrobacter aquimaris* 124	Studied cheese	
*Psychrobacter aquimaris* 129	Studied cheese	
*Psychrobacter aquimaris* 184	Studied cheese	
*Psychrobacter aquimaris* 200	Studied cheese	
*Psychrobacter cibarius* 132	Studied cheese	
*Psychrobacter cibarius* 139	Studied cheese	
*Psychrobacter cibarius* 140	Studied cheese	
*Psychrobacter cibarius* 157	Studied cheese	
*Psychrobacter cibarius* 158	Studied cheese	
*Psychrobacter cibarius* 160	Studied cheese	
*Psychrobacter cibarius* 165	Studied cheese	
*Psychrobacter cibarius* 171	Studied cheese	
*Psychrobacter cibarius* 181	Studied cheese	
*Psychrobacter cibarius* 198	Studied cheese	
*Psychrobacter aquimaris* ER15 174 BHI7	Saint-Nectaire cheese	
*Psychrobacter celer* DSM 23510	Munster cheese	
*Psychrobacter cibarius* DSM 16327	Epoisses cheese	
*Psychrobacter faecalis*	Livarot cheese	
*Psychrobacter namhaensis* 1439	Camembert cheese	

**Table 2 viruses-14-01620-t002:** PCR primers targeting specific phage genes.

Phage	Primer	Targeted CDS	Product	Sequence (5′-3′)	Annealing Temperature (°C)	Amplicon Size (bp)
Voltaire	VOLT_F	VOLT_18	Pre-neck protein	actacctaccctgcccctaa	57	705
	VOLT_R			ttcgttgaccagcacacaag		
Rousseau	ROUS_F	ROUS_20	Receptor-binding protein	ggcggttcggagggtattag	57	877
	ROUS_R			gaaccaaaccttcatcgcca		
Diderot	DID_F	DID_20	Tail tape measure protein	aaaactgctgtgactcgtgg	57	931
	DID_R			caccaaacacgccagagaaa		
D’Alembert	ALEM_F	DAL_18	RNA ligase	tggtactaatgcaggtatcggt	57	714
	ALEM_R			tcaacctcaaagcccatctct		
Montesquieu	MONT_F	MONT_53	DNA polymerase I	tgacggcaagttcaatcagc	57	683
	MONT_R			gctggttcggagtagtgtct		

**Table 3 viruses-14-01620-t003:** Morphologic characteristics of the isolated phages.

Phage	Capsid Size (nm ± SD ^1^)	Tail Size (nm ± SD)	Morphotype	Plaque Morphology
D’Alembert	88 ± 2	113 ± 2.6	myophage	Clear, small
Voltaire	47 ± 1.1	30 ± 3.8	podophage	Clear, small
Montesquieu	64 ± 1.8	184 ± 5.5	siphophage	Clear, large
Rousseau	62 ± 5.5	177 ± 15.6	siphophage	Clear, large
Diderot	57 ± 4.3	141 ± 0.9	siphophage	Clear, large

^1^ SD = Standard deviation.

**Table 4 viruses-14-01620-t004:** Host spectrum of the 5 tested phages.

Phage		Sensitive Isolates/Tested Isolates (Same Species as the Host)		Sensitive Species/Tested Species (Same Genus but Different Species as the Host)
	Propagation Strain	Isolated from the Studied Cheese	From Other Sources	
D’Alembert	*Psychrobacter aquimaris* 87	3/10	0/5	0/4
Voltaire	*Glutamicibacter arilaitensis* G65	2/13	0/1	0/3
Montesquieu	*Glutamicibacter arilaitensis* G51	7/13	0/1	0/3
Rousseau	*Brevibacterium aurantiacum* B67	1/2	0/16	0/5
Diderot	*Leuconostoc falkenbergense* 91	7/9	0/1	0/4

**Table 5 viruses-14-01620-t005:** Global metrics around sequencing and assembly steps.

Phage	Raw Reads Count		Average Size of Reads (bp)		Number of Contigs	Genome Size (kb)	Number of ORFs	Terminal Repeat Size (bp)	Best Blast Hit ^2^
	Illumina	Nanopore	Illumina	Nanopore					
Voltaire	8.9 × 10^6^	63,494	2 × 150	4320	1	18.4	26	176	*Brevibacterium* phage Cantare (83.33% id 1% cov)
Montesquieu ^1^	5.6 × 10^6^	-	2 × 150	-	1	47.7	62	-	*Arthrobacter* phage TripleJ (75.25% id 2% cov)
Rousseau	5.5 × 10^6^	15,649	2 × 150	3576	1	40.2	71	-	*Siphoviridae* sp. Isolate ctmmc7 (75.54% id 0% cov)
Diderot	6.9 × 10^6^	1,736,125	2 × 150	3074	1	27.1	40	-	*Leuconostoc* phage LN03 (98.20% id 98% cov)
D’Alembert	7.1 × 10^6^	115,763	2 × 150	9872	1	92.5	158	5719	*Vibrio* phage vB_VhaM_VH-8 (83.95% id 34% cov)

^1^ As explained in Section 2, Montesquieu genome assembly was obtained from Illumina reads only. ^2^ Accession numbers for each related phage: *Brevibacterium* phage Cantare: MK016493; *Siphoviridae* sp. Isolate ctmmc7: BK019734; *Leuconostoc* phage PhiLN03: NC_024390; *Vibrio* phage vB_VhaM_VH-8: MN497415; *Arthrobacter* phage TripleJ: MN234178.

**Table 6 viruses-14-01620-t006:** Completeness and encapsidation strategy for the five phages.

	Illumina Only Assembly			Final assembly	
	Size in bp	Terminal Repeat	PhageTerm Prediction: Boundaries and Encapsidation Strategy	Size in bp after Polishing	Terminal Repeat
Voltaire	18,300	/	Redundant, permuted and unknown ^1^	18,418	176 bps ITR
Montesquieu	47,703	127 bp DTR, assembly artefact removed	Headful (pac)	47,576	/
Rousseau	40,294	127 bp DTR, assembly artefact removed	Cos (3′)	40,167	/
Diderot	/	/	Cos (3′)	27,116	/
D’Alembert	86,864	127 bp DTR, assembly artefact removed	5719 bps DTR (long)	92,456	5719 bps DTR

^1^ phi29-like phage packaging strategy not predictable using PhageTerm [44].

## Data Availability

The genomic data for this study have been deposited in the European Nucleotide Archive (ENA) at EMBL-EBI under accession number PRJEB48484. Assembled/annotated genome sequences of the five isolated phages were precisely deposited under the accession numbers OV696617 (Voltaire), OV696619 (Montesquieu), OV696620 (D’Alembert), OV696621 (Diderot) and OV696622 (Rousseau).

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
