# Peer review of "Virulent Phages Isolated from a Smear-Ripened Cheese Are Also Detected in Reservoirs of the Cheese Factory"

_viruses, 2022, doi:10.3390/v14081620_

Round 1

Reviewer 1 Report

The paper by Paillet et al analysed the rind of smear-ripened cheeses and identified 5 virulent phages, of which 4 are likely to be new phages.

The author reported that all phages exhibit a narrow host range. It would be useful if the authors could at least speculate on why this is the case.

The author performed the sequence analysis of all phages isolated using Nanopore and Illumina sequencing. Some annotations have also been attempted. However, it would be crucial for the author to also analyse the sequenced genome for virulence factors. There is no doubt that the phage-encoded virulence factors will have huge implications in the food industry.

Finally, it would be good if the author can comment on the stability of the isolated phages. Stability assays (at different temperatures and pH) are relatively easy experiments to perform but will significantly improve the quality of the paper.

Author Response

The paper by Paillet et al analysed the rind of smear-ripened cheeses and identified 5 virulent phages, of which 4 are likely to be new phages.

The author reported that all phages exhibit a narrow host range. It would be useful if the authors could at least speculate on why this is the case.

Answer: According to the literature (e.g. Weinbauer 2004), a narrow host range is generally expected for most phages. Polyvalent phages have been isolated and described, of course, but it seems that they are quite rare in nature. We discussed our results in this way within the discussion section.

The author performed the sequence analysis of all phages isolated using Nanopore and Illumina sequencing. Some annotations have also been attempted. However, it would be crucial for the author to also analyse the sequenced genome for virulence factors. There is no doubt that the phage-encoded virulence factors will have huge implications in the food industry.

Answer: we followed the recommendation of the reviewer by comparing (BlastX) the CDSs of the five newly sequenced phages to Virulence Factor DataBase (VFDB, http://www.mgc.ac.cn/VFs/) and found no virulence genes. Although not suggested by the reviewer, we also searched for antibiotic resistance genes using Resfinder (https://cge.cbs.dtu.dk/services/ResFinder/) and found no ARGs. Those information were added to the manuscript (Material and Methods and Results sections).

Finally, it would be good if the author can comment on the stability of the isolated phages. Stability assays (at different temperatures and pH) are relatively easy experiments to perform but will significantly improve the quality of the paper.

Answer: We would like to thank the reviewer for his inspiring comment. We did not completely followed his suggestion. However, from a technological point of view (in cheese production) we found very interesting and relevant to test whether phage infection was successful in the range of temperature compatible with the process (from 12°C to 30°C). We therefore performed an additional experiment and added the corresponding description in the material and methods section and in the results (with a supplementary figure).

Reviewer 2 Report

The present paper appropriately addresses the finding of some new bacteriophages from a special French cheese. It is well presented and the conclusions are fine.

I believe that the paper should be published in its present form since it may be of utility not only for the cheesemakers and the Food Science in general but also for those bacteriophage-specialized virologists. 

Author Response

The present paper appropriately addresses the finding of some new bacteriophages from a special French cheese. It is well presented and the conclusions are fine.

I believe that the paper should be published in its present form since it may be of utility not only for the cheesemakers and the Food Science in general but also for those bacteriophage-specialized virologists. 

Answer: we thank the reviewer for this nice comment.

Reviewer 3 Report

The manuscript “Virulent phages isolated form a smear-ripened cheese are also detected in reservoirs of the cheese factory” aims to investigate the phage content of the rind from smear-ripened cheeses, in addition to identifying the source of such phage in the production environment. This substantial body of work provides significant contributions to the field though the isolation of, and detailed genetic characterisation of, at least four novel phages which infect the lesser studied bacterial species utilised in cheese production. The study also identifies specific locations within the production facility which serve as persistent phage reservoirs, thus highlighting critical points of infection which will be of interest to the broader audience. In addition, the application of ViPTree for taxonomic classification of the novel isolates is to be commended, as indeed are the high quality TEM images provided within.

Specific comments and questions are provided below for the authors.

L25 - The authors make no reference in the abstract as to the isolation and identification of bacterial species from the samples. As this represents a major part of the study it should be included to reflect the body of work undertaken.

L53 - needs clarification.

L58 - suggest “bacterial species” in place of “microorganisms” for clarity

L59 - LAB – please provide a more rounded explanation in the first instance i.e., species used for milk acidification? Function of LAB in addition to microbial safety and preservation?

L60-63 - hurdle effect – please provide an explanation for readers who may not be aware of the concept. Possibly introduce at L52?

L74 - suggest moving this paragraph to the start of the introduction to give the reader an early introduction to smear ripened cheeses

L90 – suggest including the number of phage/host combinations isolated and investigated as part of the study

L97 - samples – can further detail be provided i.e., type of cheese? production date? were all samples of the same type?

L105 - can a dilution range be provided?

L112 - why are two different temperatures used for the isolation of total aerobic bacteria (28 oC) vs. anaerobic LAB (30 oC)?

Sections 2.2.1 and 2.2.2 – suggest merging to a single section for enumeration and isolation as there is some method duplication

L136 - how were bacterial cultures maintained following isolation? Plates / glycerol stocks?

L168 - SSU – please explain in first instance

L168 - please indicate the polymerase used and expected amplicon size

L192 - is there a reason for differing volumes (20ml vs. 9ml) between strains?

L192 - 10 µl of sample seems low for enrichment? Was full lysis observed following overnight incubation?

L200 - please provide a reference for double layer method

L203 - was the culture volume used dependent on the strain to be tested or another condition?

L212 - were any variances in plaque morphology observed on a single strain?

L214 - how was the titre of the pure phages determined?

L216 - what volumes of phage / host were used to obtain confluent lysis?  

L221 - as for L200

L228 – suggest removing the note on personal communication from Nicolas Ginet and moving it to Acknowledgements

L330 - suggest replacing “when available” to “where possible”

L335 - Table 1 – for the studied cheese isolates – was there an indication that the individual isolates were non-clonal? What was the criteria for selection other than being of the same species or was selection randomised?

L350 - L354 - suggest moving this to L345/6 for clarity on sample collection methods and then describing the precipitation method to avoid repetition.  What was the volume of the liquid samples obtained for testing?

L358 - why not test additional strains if available? Could novel or emerging phage have been missed by restricting the screening panel to only known sensitive strains?  Perhaps a point for the discussion?

L359 - suggest changing “virome sample” to “production sample” throughout

L360 - refer to Table 1 to remind reader of strain/species detail

L362 - see suggestion from L359

L362 - method for enrichment already described earlier in the text

No line numbers visible after this point in the manuscript - will refer to sections 

Section 3.1

suggest presenting a Table with overview of species/strains/ % distribution etc. if possible. Was there a variance in counts / species distribution between the individual cheese samples? Can additional detail be provided?

203 bacterial isolates obtained. How were strains in Table 1 selected for host range analysis?

Section 3.2

un-necessary repeat of methods. Detail on plaque morphology? Suggest providing a Table with phage isolate details if possible

States that “rind viromes” were pooled, this is not clear in the methods section?  Were samples pooled by type/date etc?

Methods state two rounds of plaque purification?

Do the authors have any thoughts on why phage were not isolated for these three frequently isolated species?  

Section 3.2.2

Table 4 – please include a column indicating the host strain to clarify for the reader. Is the observed narrow host range in line with literature for phage of these species or is this known?

Section 3.2.3

After Table 6 – What is distinct about the morphological observations of phage Volatire? Although interesting, suggest removing this statement in relation to taxonomy.

Fig4. Were RBPs identifiable for all phage isolates in this study? Only indicated in figure for phages Rousseau and Diderot.

Section 3.2.4

were phage titres of factory samples determined? Impact of burden on cheese production if so?

Typo - Salting plaques?

PCR results indicate multiple phage – was this investigated? Was the presence of prophage in the host genome(s) considered?

Section 4

please provide a reference for lysogens/ temperate phage

PCR detected the phages with some nucleotide variance was observed. Could this indicate evolution and host adaptation over time?

Author Response

The manuscript “Virulent phages isolated form a smear-ripened cheese are also detected in reservoirs of the cheese factory” aims to investigate the phage content of the rind from smear-ripened cheeses, in addition to identifying the source of such phage in the production environment. This substantial body of work provides significant contributions to the field though the isolation of, and detailed genetic characterisation of, at least four novel phages which infect the lesser studied bacterial species utilised in cheese production. The study also identifies specific locations within the production facility which serve as persistent phage reservoirs, thus highlighting critical points of infection which will be of interest to the broader audience. In addition, the application of ViPTree for taxonomic classification of the novel isolates is to be commended, as indeed are the high quality TEM images provided within.

Specific comments and questions are provided below for the authors.

L25 - The authors make no reference in the abstract as to the isolation and identification of bacterial species from the samples. As this represents a major part of the study it should be included to reflect the body of work undertaken.

Answer: we modified the abstract according to the reviewer comment.

L53 - needs clarification.

Answer: we deleted the sentence since we did not want to develop much the safety aspect which is not closely related to the main topic of the article.

L58 - suggest “bacterial species” in place of “microorganisms” for clarity

Answer:  Pasteurization may also inactivate yeasts and moulds, which is why we decided to keep the broader term “microorganisms” here.  

L59 - LAB – please provide a more rounded explanation in the first instance i.e., species used for milk acidification? Function of LAB in addition to microbial safety and preservation?

Answer: as suggested by the reviewer, we provided more details regarding the function of LAB.

L60-63 - hurdle effect – please provide an explanation for readers who may not be aware of the concept. Possibly introduce at L52?

Answer: we provided a definition of this term as suggested by the reviewer.

L74 - suggest moving this paragraph to the start of the introduction to give the reader an early introduction to smear ripened cheeses

Answer: we tried to take into account the reviewer comment but, in our opinion, it is easier to follow if we keep this paragraph after the general description of the problematic of bacteriophages in the dairy industry.    

L90 – suggest including the number of phage/host combinations isolated and investigated as part of the study

Answer:  we modified the description of the study at the end of the introduction in order to take into account the comment of the reviewer.

L97 - samples – can further detail be provided i.e., type of cheese? production date? were all samples of the same type?

Answer: we modified the sentence accordingly.

L105 - can a dilution range be provided?

Answer: we added this information.

L112 - why are two different temperatures used for the isolation of total aerobic bacteria (28 oC) vs. anaerobic LAB (30 oC)?

Answer: by experience in our lab, we usually incubate plates for the count of aerobic bacteria at 28°C since most cheese surface bacteria can grow fast at this temperature and because some psychrophilic bacteria are not able to grow at higher temperature. For LAB, the optimal growth temperature of many strains is higher than 30 °C but this temperature is suitable for the development of both mesophilic and thermophilic LAB.

Sections 2.2.1 and 2.2.2 – suggest merging to a single section for enumeration and isolation as there is some method duplication

Answer: we followed the suggestion.

L136 - how were bacterial cultures maintained following isolation? Plates / glycerol stocks?

Answer: glycerol stocks after isolation and stored at -80°C. We added this information.

 L168 - SSU – please explain in first instance

Answer: we added the information

L168 - please indicate the polymerase used and expected amplicon size

Answer: we added the information

L192 - is there a reason for differing volumes (20ml vs. 9ml) between strains?

Answer: Aerobic strains were cultivated into 100 ml flasks containing 20 ml of broth medium under agitation whereas lactic acid bacteria were cultivated in culture tubes almost completely filled with MRS broth without agitation to limit oxygenation. We thus adapted the medium volume to the size of our material.

L192 - 10 µl of sample seems low for enrichment? Was full lysis observed following overnight incubation?

Answer: It is true that the dilution factor is high, but to achieve a good enrichment, we want the phage to achieve several lysis cycles. Yet if the phage is in excess and kills directly the bacterial host, the enrichment will be less effective.

L200 - please provide a reference for double layer method

Answer: we added the reference

L203 - was the culture volume used dependent on the strain to be tested or another condition?

Answer: yes, actually some strains were growing more slowly than the others. For such strains we thus decided to inoculate a larger volume of the overnight culture into the molten top agarose to ensure that the whole plate was covered by a bacterial lawn.

L212 - were any variances in plaque morphology observed on a single strain?

Answer: no we didn’t observe different plaque morphology on a single strain.

L214 - how was the titre of the pure phages determined?

Answer: this was explained later (titre of large phage stocks) but we moved the paragraph here.

L216 - what volumes of phage / host were used to obtain confluent lysis? 

Answer: we added the details

L221 - as for L200

Answer: we added the reference

L228 – suggest removing the note on personal communication from Nicolas Ginet and moving it to Acknowledgements

Answer: done.

L330 - suggest replacing “when available” to “where possible”

Answer: done.

L335 - Table 1 – for the studied cheese isolates – was there an indication that the individual isolates were non-clonal? What was the criteria for selection other than being of the same species or was selection randomised?

Answer: we had no indication that the individual isolates were non-clonal, we randomly selected up to 13 isolates per species (where possible) in order to perform the spot assays. We added this information.

L350 - L354 - suggest moving this to L345/6 for clarity on sample collection methods and then describing the precipitation method to avoid repetition.  What was the volume of the liquid samples obtained for testing?

Answer: we made the modification suggested by the reviewer. The volume of liquid samples analysed was 50 ml, we also added this information.

L358 - why not test additional strains if available? Could novel or emerging phage have been missed by restricting the screening panel to only known sensitive strains?  Perhaps a point for the discussion?

Answer: the objective of this experiment was simply to identify the sources of the five phages newly described in the study. Of course, it would have been interesting to test the whole collection of bacterial isolates but the objective of such experiment is quite different. 

L359 - suggest changing “virome sample” to “production sample” throughout

Answer: we replaced “virome sample” by “viral extract from the production sample”

L360 - refer to Table 1 to remind reader of strain/species detail

Answer: done.

L362 - see suggestion from L359

Answer: same answer.

L362 - method for enrichment already described earlier in the text

Answer: the reviewer is right, the enrichment procedure was described earlier (phage isolation section) but we believe that it is important to describe it again here (not as detailed as in the previous section) to fully understand our workflow.

No line numbers visible after this point in the manuscript - will refer to sections 

Section 3.1

suggest presenting a Table with overview of species/strains/ % distribution etc. if possible. Was there a variance in counts / species distribution between the individual cheese samples? Can additional detail be provided?

Answer: unfortunately, as mentioned in the material and methods, an initial selection of apparently different morphotypes was performed based on colony morphology. The procedure to construct our collection of isolates was therefore not random. Giving indications such as those proposed by the reviewer could thus provide a biased overview of the bacterial composition of the studied cheese. For this reason, we decided to not address this comment.

203 bacterial isolates obtained. How were strains in Table 1 selected for host range analysis?

Answer: we randomly selected the bacterial isolates for each bacterial species. We added this information in the material and methods section.

Section 3.2

un-necessary repeat of methods. Detail on plaque morphology? Suggest providing a Table with phage isolate details if possible

Answer: according to the reviewer suggestion, we deleted the methods and added a column about plaque morphology in table 3.

States that “rind viromes” were pooled, this is not clear in the methods section?  Were samples pooled by type/date etc?

Answer: the three cheeses were replicates of the same batch (three cheeses from the same producer, produced at the same date). Since it was already mentioned in the methods section, we deleted this information in the results.

Methods state two rounds of plaque purification?

Answer: the first observation of lysis plaque during the spot-assay constitutes the fisrt round of purification and we then performed two additional rounds (thus 3 in total).

Do the authors have any thoughts on why phage were not isolated for these three frequently isolated species? 

Answer: no we don’t, several hypotheses can explain this result but we have no indication to favour one more than the others. Those species (or strains) may be resistant to phages in general, phages for those species (strains) might not be present in the sampled cheese at the sampling date, phages for those species (strains) might not form lysis plaques under our culture conditions, we might have only picked resistant clones of the corresponding species (strains).

Section 3.2.2

Table 4 – please include a column indicating the host strain to clarify for the reader. Is the observed narrow host range in line with literature for phage of these species or is this known?

Answer: we added a column with the propagation strain as suggested by the reviewer. The observed narrow host range is common, we commented that point within the discussion section.

Section 3.2.3

After Table 6 – What is distinct about the morphological observations of phage Volatire? Although interesting, suggest removing this statement in relation to taxonomy.

Answer: we deleted the reference to the morphological observations of Voltaire in relation to taxonomy.

Fig4. Were RBPs identifiable for all phage isolates in this study? Only indicated in figure for phages Rousseau and Diderot.

Answer: RBPs were only identified in Rousseau and Diderot, not in the three other phages.

Section 3.2.4

were phage titres of factory samples determined? Impact of burden on cheese production if so?

Answer: no unfortunately we didn’t determine phage titres on those samples.

Typo - Salting plaques?

Answer: we replaced “salting plaques” by “salting tables”

PCR results indicate multiple phage – was this investigated? Was the presence of prophage in the host genome(s) considered?

Answer: this result indeed suggest that multiple closely-related Leuconostoc phages co-occur in the dairy plant. We believe that further investigations are now required (including isolation of representative phages and genome sequencing of the host to look for the presence of prophages) but feel that a dedicated study of Leuconostoc phages should be designed to explain this observation.   

Section 4

please provide a reference for lysogens/ temperate phage

Answer: We added the reference

PCR detected the phages with some nucleotide variance was observed. Could this indicate evolution and host adaptation over time?

Answer: we added a sentence within the discussion section in order to comment this result.

Round 2

Reviewer 3 Report

The manuscript “Virulent phages isolated form a smear-ripened cheese are also detected in reservoirs of the cheese factory” aims to investigate the phage content of the rind from smear-ripened cheeses, in addition to identifying the source of such phage in the production environment. This substantial body of work provides significant contributions to the field though the isolation of, and detailed genetic characterisation of, at least four novel phages which infect the lesser studied bacterial species utilised in cheese production. The study also identifies specific locations within the production facility which serve as persistent phage reservoirs, thus highlighting critical points of infection which will be of interest to the broader audience. In addition, the application of ViPTree for taxonomic classification of the novel isolates is to be commended, as indeed are the high quality TEM images provided within.